# Characterization of the fungal community in the canopy air of the invasive plant *Ageratina adenophora* and its potential to cause plant diseases

Lin Chen[1,2,3], Kai Fang[1,2,3], Xing-Fan Dong[2,3], Ai-Ling Yang[2,3], Yu-Xuan Li[2,3], Han-Bo Zhang ORCID [1,2,3] *

**1** School of Ecology and Environmental Science, Yunnan University, Kunming, Yunnan Province, China, **2** State Key Laboratory for Conservation and Utilization of Bio-Resources in Yunnan, Yunnan University, Kunming, Yunnan Province, China, **3** School of Life Sciences, Yunnan University, Kunming, Yunnan Province, China

* zhhb@ynu.edu.cn

**Data Availability Statement:** All relevant data are within the paper and its Supporting Information files. All sequences used in this article were uploaded to the NCBI database under accession

## Abstract

Airborne fungi and their ecological functions have been largely ignored in plant invasions. In this study, high-throughput sequencing technology was used to characterize the airborne fungi in the canopy air of the invasive weed *Ageratina adenophora*. Then, representative phytopathogenic strains were isolated from *A. adenophora* leaf spots and their virulence to *A.adenophora* as well as common native plants in the invaded range was tested. The fungal alpha diversities were not different between the sampling sites or between the high/low part of the canopy air, but fungal co-occurrences were less common in the high than in the low part of the canopy air. Interestingly, we found that the phytopathogenic Didymellaceae fungi co-occurred more frequently with themselves than with other fungi. Disease experiments indicated that all 5 Didymellaceae strains could infect *A. adenophora* as well as the 16 tested native plants and that there was large variation in the virulence and host range. Our data suggested that the diverse pathogens in the canopy air might be a disease infection source that weakens the competition of invasive weeds, a novel phenomenon that remains to be explored in other invasive plants.

## Introduction

It has been suggested that invasive plants have the ability to decrease biodiversity in local eco-systems through competition [1]. The enemy release hypothesis (ERH) partially explains this competitive advantage, as hosts in the introduced range escape from their enemies [2–7]. Klir-onomos (2002) showed that five of North America's most damaging exotic plant invaders modified the soil microbial community in ways that benefit themselves (i.e., positive feedback) [8]. Callaway *et al.* (2004) found a switch from negative to positive plant-soil feedback for spot-ted knapweed when moving from its native to its exotic range [9]. However, some invasive

Airborne fungi associated with an invasive plant

numbers PRJNA590841 (the next-gen data), and MK813969 - MK814043 (cultured isolate data).

**Funding:** This work was funded by the National Natural Science Foundation of China (grant Nos. 31770585, 31360153). H-BZ received the awards.

**Competing interests:** The authors have declared that no competing interests exist.

plants have been shown to accumulate pathogens in their invaded ranges, which is called the pathogen accumulation and invasive decline (PAID) hypothesis [10]. In some cases, pathogen accumulation can limit the spread of invasion [11]. In eastern North America, the highly invasive annual grass *Microstegium vimineum* becomes significant declines in growth performance in natural populations due to accumulate *Bipolaris* species and other fungal pathogens [12, 13].

Besides the direct adverse impacts on invasive host, these accumulated pathogens can indirectly exacerbate the effects of invasions if they are transmitted in the invaded ecosystem and affect the susceptible of native hosts [34], which is defined as disease-mediated invasion (DMI) [14, 15]. Alternatively, invasive plants could be a reservoir of local pathogens, which may spread to wild plants, crops and generate very large ecological risks [10, 16–18]. In the UK, the invasive *Rhododendron ponticum* is a key foliar reservoir host for the native pathogens *Phytophthora ramorum* and *P. kernoviae* [19]. Evaluating these impacts of invasive hosts in the introduced range largely depends on the understanding if these fungi can be released from hosts into the surrounding air and if they have the potentials to cause diseases on neighboring native plants.

Airborne fungal propagules have been a major focus of epidemiological studies in traditional plant pathology for native plants and crops. For example, most airborne spores come from fungi growing on living or dead plants [20], and their abundance is influenced by multiple abiotic factors, such as temperature [21], humidity [22], rainfall [23], and wind speed [24]. Scherm & van Bruggen (1995) reported that lesions on source plants infected with an isolate of *Bremia lactucae* sporulated at night and released spores beginning at sunrise [25]. Plant canopy air, with complex microclimates, thus acts as an important space for dynamic host-pathogen interactions [26]. Pathogens in the canopy air can infect plants under optimal microclimate conditions and in turn, release their spores into the air to complete their life cycle; meanwhile, the host can change the canopy microclimate to affect pathogen spread and disease development [26]. Some studies for the monoculture crop also have shown that numbers of conidia collected decreased with increasing height within and above the crop canopy [27–29]. Similarly, for the monoculture invasive plant, its canopy air would harbor diverse fungal spores released from leaf spots, which in turn, may infect host itself as well as neighboring species. It should be expected that: (1) The closer to the plant host, the higher diversity and more complicated co-occurring connections of fungi; (2) there are the sharing between fungi in the canopy air of invasive plants and host-associated pathogenic fungi. Currently, there is a total lack of the knowledge of the airborne phytopathogenic fungi surrounding the invasive plants.

The invasive weed *Ageratina adenophora* is one of the most serious invasive weeds in Yunnan Province, China. It has rapidly spread in southwestern China and is expected to invade southern and south-central China unless it is controlled [30]. Recently, several studies indicated that *A. adenophora* could be infected by the foliar fungal pathogens *Phaeoramularia sp.* [31], *Passalora ageratinae* [32], *Alternaria alternata* [33], and *Collectotrichum* [34,35]. In recent, our group has indicated that the foliar fungi from family Didymellaceae are adverse to the growth of *A. adenophora* [36], and these fungi frequently occurs in the surrounding environment, such as in the withered leaves and the canopy air of hosts [37]. In this study, two experiments were designed to verify the expectation above. Firstly, we used the high-throughput sequencing technology to characterize the fungal community and analyze their co-occurrence network in the canopy air of *A. adenophora* (experiment 1); and then the shared representative phytopathogenic fungi from family Didymellaceae were isolated from leaf spots of *A. adenophora* and their virulence to *A. adenophora* as well as native plants was tested (experiment 2).

## Materials and methods

### Experiment 1: High-throughput sequencing of canopy fungal communities

**Collection of airborne fungal spores.**   The samples were collected in two regions in which *A. adenophora* heavily occurs in Yunnan Province, China. These sampling sites are not located within the protected area and no sampling permit is required. At 2–4 pm, April 10 and 15, 2018, three sites per region were selected to collect the air in the same day and considered as the biological duplication. In each sampling site, we sampled two heights in the air column, and totally we collected 12 air samples. Briefly, in each region, three sampling sites over 1 km apart were randomly selected (S1 Table, sampling sites information). The canopy air in each site was sampled from both the immediate canopy (the air from the low part of the canopy, defined as LC) and 1.5 meters above the vegetation (the air from the high part of the canopy, defined as HC). For each sample, airborne spores were concentrated from 1000 liters of air (~5mins), using a 9cm diameter Petri dish filled with a sterilized cellulose acetate membrane by a surface air system (SAS) Super ISO 180 (VWR International PBI SRL, San Giusto, Italy). The Petri dishes were immediately sealed with Parafilm® "M" (Pechiney Plastic Packaging, Menasha, WI) after air collection and brought back to the laboratory. Then, the cellulose acetate membrane was removed, cut into small pieces and placed in 2 mL centrifuge tubes. The fungal spores on the cellulose acetate membrane were oscillated and removed with 600 μL sterile water and sterilized steel balls in a magnetic shaker for 10 minutes. The cellulose acetate membrane and steel balls were removed from the centrifuge tubes. Then, the spores from each sample were centrifuged and pooled, and the precipitant was stored in a -4˚C refrigerator prior to DNA extraction.

**DNA extraction.**   Prior to DNA extraction, the concentrated spores were resuspended in approximately 250 μL of sterile distilled water. The resuspended fungal spores were extracted using the commercial DNA extraction kit FastDNA® SPIN Kit for Soil (MP Biomedicals, Irvine, CA). The DNA quality was monitored by 0.8% agarose gel electrophoresis. The extracted DNA was diluted to a concentration of 2 ng/μL to balance the DNA amount among the samples, and then stored at -20˚C until further processing. Meanwhile, the sterilized cellulose acetate membranes without performing collection of air spores were used as the extraction blanks, which were failed in DNA extraction and were excluded from the further processing.

The diluted DNA was used as a template for PCR amplification. For the fungal diversity analysis, the fungal rRNA gene was amplified with primers ITS1-1737F (`GGA AGT AAA AGT CGT AAC AAGG`) and ITS2-2043R (`GCT GCG TTC ATC GAT GC`) targeting ITS1-ITS2 (approximately 246 bp) [38,39]. All the amplifications were conducted in an ABI GeneAmp® 9700 (Applied Biosystems Inc., Foster City, CA) at 95˚C for 3 min, followed by 37 cycles at 95˚C for 30 s, 55˚C for 30 s, and 72˚C for 45 s and a final elongation step at 72˚C for 10 min. The PCRs contained 2 μL 10 × PCR buffer, 2 μL 2.5 mM dNTPs, 0.8 μL 5 μM ITS1-1737F and ITS2-2043R primer, 0.2 μL of TakaRa rTaq DNA polymerase and 10 ng DNA template, with a final volume of 20 μL. Equal amounts of purified amplicon were pooled for subsequent sequencing using the Miseq sequencing platform at Shanghai Majorbio Biopharm Technology Co., Ltd. (Shanghai, China). The next-gen data were submitted to GenBank under bioproject accession numbers PRJNA590841.

**Bioinformatic analysis of the high-throughput sequencing.**   The raw sequencing data were in the FASTQ format. The paired-end reads were preprocessed using Trimmomatic software [40] to detect and remove ambiguous bases (N). We also removed low-quality sequences, i.e., those with an average quality score below 20, using the sliding window trimming approach. After trimming, the paired-end reads were assembled using FLASH software [41]. The parameters of assembly were a minimum of 10 bp overlapping, a maximum of 200 bp

overlapping and a 20% maximum mismatch rate. The sequences were further denoised as follows: the reads with ambiguous, homologous sequences or sequences below 200 bp were abandoned, reads with 75% of bases above Q20 were retained, and reads with chimeras were detected and removed. These steps were performed using QIIME software (version 1.8.0) [42].

The clean reads were subjected to primer sequence removal and clustering to generate operational taxonomic units (OTUs) using UPARSE software with a 97% similarity cutoff [43]. The representative read of each OTU was selected using the QIIME package. All the representative reads were annotated and blasted against the UNITE database using BLAST [44,45].

**Data analysis of the high-throughput sequencing.**   After OTU classification, we first deleted the unidentified OTUs at the phylum level and then subsampled all samples to the 30376 reads (minimized reads' sample) using the "sub.sample" function in MOTHUR v1.35.1. All the data were analyzed except those that were removed during subsampling (see S2 Table Data Subsampled OTUs table). The rarefaction curve for each sample was calculated using MOTHUR "rarefaction.single" and plotted using Graphpad Prism v7 (GraphPad Software, Inc., CA, USA). The alpha diversity was calculated using the R package "vegan" and plotted using Graphpad Prism v7. The alpha fungal diversity between two regions, as well as between two heights in the air column was compared using paired t test by SPSS 25.0 (IBM, NY, USA). We used two methods to compare the diversity between two regions, i.e. considering the heights in the air column (LC: 3 vs 3, HC: 3 vs 3) and not considering the height in the air column (6 vs 6). The same method was used to compare the different height in the air column with and without considering the region. Similarly, the differences in fungal abundance between two regions, as well as between two heights in the air column were also compared using paired t test using the methods above described at phylum, family, and genus level, respectively. The fungi with an average abundance of less than 0.5% were excluded from the comparison. For the phylogenetic analysis of Didymellaceae, the most similar sequences were downloaded from NCBI and neighbor joining (NJ) was used to construct phylogenetic trees with Mega X using 18S rRNA gene sequences (approximately 220 bp). *Leptosphaeria conoidea* (CBS 616.75) and *L. doliolum* (CBS 505.75) were selected as the outgroup. The node stability was assessed with 1,000 bootstrap replicates. The relative abundance of Didymellaceae fungi is shown in a heatmap plotted by the R package "pheatmap".

Co-occurrence patterns do not allow mapping of microbial interactions directly, but provide information on particular groups sharing habitats or performing similar ecological functions [46]. Therefore, to reveal the co-occurrence among airborne fungi, network analyses of the OTU matrix based on Spearman's Rho were calculated by the "psych" R package. The OTUs that occurred only in one sample were excluded when structuring the co-occurrence networks. We further divided the synthetic dataset into two parts: LC and HC of *A. adenophora*. The co-occurrence among all the fungi was calculated for both LC and HC. These two valid co-occurrence events were considered to be robust if the correlation coefficient $\rho > |0.8|$ and if they were statistically significant at $P < 0.05$ [46–49]. The network visualization was generated with Gephi v 0.9.2 [50]. Other graphs were plotted by GraphPad Prism v7.

## Experiment 2: Isolation of leaf spot fungi and performance of the disease experiment

**Isolation and molecular identification of leaf spot fungi.**   To verify if there is fungal sharing of the leaf spots with the canopy air of *A. adenophora*, we selected one population of *A. adenophora* from one site (KM3) to perform fungal isolation. Diseased leaves with morphologically different symptoms were collected. Healthy leaf tissues and the margins of diseased

tissues of each leaf spot were cut into six sections of 6 mm$^2$ and surface sterilized. The disinfected fragments were then plated onto PDA and incubated at ambient temperature for 6–8 days or until mycelia growing from the leaf fragments were observed. Then, all the fungal colonies grown from the leaf fragments were purified and used to determine the phylogenetic position and in the disease experiments. All the fungi were maintained as pure cultures at Yunnan University (Kunming, China). All the fungi were sequenced by the internal transcribed spacer (ITS) region using primers ITS 1 (`TCC GTA GGT GAA CCT GCGG`) and ITS 4 (`TCC GCT TAT TGA TAT GC`) [51–52]. Sequences of pure isolates were clustered to generate OTUs using MOTHUR software with a 100% similarity cutoff [43]. These cultured OTUs were named as cOTUs to distinguish from the OTUs generated by next-generation sequencing. The fungi were annotated and blasted against the UNITE database [45] using the MOTHUR classifier (confidence threshold of 80%) [44]. The nucleotide sequences of these OTUs have been deposited in GenBank under accession numbers MK813969—MK814043.

**Identifying the shared OTUs of leaf spot fungi with airborne fungi of *A. adenophora*.** To identify the fungal sharing, the ITS sequences from each representative OTU obtained from airborne fungi were selected to perform alignment with those sequences obtained from cultural leaf spot fungi above. Because the ITS sequence obtained by the high-throughput sequencing technology was short (~250bp), the alignment was trimmed to this range and was clustered to generate OTUs again using MOTHUR software with a 97% similarity cutoff [43]. Deletions/insertions were considered when comparing the sequences from next-gen and cultures. Those OTUs clustered from both sources were defined as the shared fungi; the Venn and bubble diagrams were used to show the shared fungal OTUs. Then, the shared Didymellaceae fungi obtained from leaf spots were selected to test their pathogenicity in the disease experiment. The used multiple isolates from one OTU were obtained from different leaf spots to avoid the origin of the same clone.

**Disease experiment.** The disease experiment was performed in the field. The field site is located in Xishan mountain, Kunming (Lat 24°58′24″ N and Lon 102°37′17″ E) and has an elevation of 2,214 m. The site is not located within the protected area and no sampling permit is required. In this site, there is a natural plant assemblage, which includes weeds, forbs, trees and vines, native to Yunnan. The experimental period extended from June to the end of October 2018, the primary growth season for plants in Kunming. In total, *A. adenophora* and 16 other native plants (*Ampelopsis sinica*, *Zehneria maysorensis*, *Reinwardtia indica*, *Fallopia multiflora*, *Pharbitis nil*, *Rubia cordifolia*, *Arthraxon hispidus*, *Urena lobata*, *Abelmoschus moschatus*, *Achyranthes bidentata*, *Quercus glauca*, *Lindera communis*, *Celtis tetrandra*, *Betula alnoides*, *Smilax scobinicaulis*, and *Pueraria peduncularis*) were tested. Because necrotrophs infect frequently through wounds [53], our experiment was performed as previously reported to test the virulence of necrotrophs in tropical forests [54]. Briefly, the fungi were grown on PDA for 7 days, and 6mm$^2$ fungal mycelium agar dishes were obtained and used to inoculate in plants in the field. Mature and healthy leaves were selected for inoculation. Small wounds were made by lightly touching the underside of the leaf with toothpicks; this resulted in 7 pin pricks in an area of 0.5 cm$^2$. The inoculum agar was pressed against the wound using cellulose tape on the underside of the leaf and clipped in place with a bent hair clip. The wounds were labeled with the strain number. One week after the inoculation, the leaves were harvested and leaf spot size was measured. The leaf spot size for each strain was visualized by a heat map, which was plotted by the R package "pheatmap". The PDA culture agar without fungus was used as the negative control. In the case that the obvious symptom was developed in the negative control group, the same batch of the fungi was tested again.

## Results

### Experiment 1: High-throughput sequencing of canopy fungal communities

**The fungal diversity and species in the canopy air.**   After the removal of the ambiguous OTUs and subsampling, a matrix with 12 samples × 178 OTUs was obtained, and all the samples reached saturation at ~15,000 sequences (see S1 Fig rarefaction curves). The alpha diversities, including the Shannon index and species richness, were different neither between the regions (with or without considering the height in the air column) nor between the high (HC) and low (LC) parts of the canopy (with or without considering the region) (Fig 1, P > 0.05, paired t test), but there was a large variation within sites.

The fungi belonged to 3 phyla, 59 families and 85 genera. Fungi from the Ascomycota accounted for 98.86% of the overall abundance and included Cladosporiaceae (61.21%), Didymellaceae (5.22%), Hypocreales (4.63%), Cordycipitaceae (3.84%), Nectriaceae (3.21%), Pleosporaceae (2.52%), Mycosphaerellaceae (2.26%) and Aspergillaceae (1.84%). The dominant genera are *Cladosporium* (60.58%), *Sarocladium* (4.42%), *Lecanicillium* (3.82%), *Alternaria* (2.08%), *Epicoccum* (1.49%), *Penicillium* (1.03%), and *Fusarium* (0.53%) (Fig 2). The fungal community composition showed no significant difference between the LC and HC canopy air and between two regions (P > 0.05, paired t test), with the exception that *Epicoccum* varied significantly between YL and KM when no considering the LC and HC (P = 0.028, paired t test).

**Fungal co-occurrence network analysis.**   In total, the fungal co-occurrence network had 62 nodes and 85 edges, with 20 negative edges (Fig 3A and 3B). The air from the low part of the canopy (LC) had more fungal co-occurrences than those of the air from the high part of the canopy (HC). The most abundant *Cladosporium* rarely linked among themselves or with other fungi positively. In most cases, common phytopathogens, e.g., *Alternaria* (OTU117) and *Fusarium* (OTU202), are positively linked with other fungi (Fig 3A). Interestingly, we found that diverse fungi belonging to the Didymellaceae family frequently linked with each other positively, e.g., OTU11-OTU43, OTU43-OTU295, and OTU97-OTU295, but linked with other fungi negatively, e.g., OTU97 (Didymellaceae) with OTU117 (*Alternaria*), and with OTU325 (*Fusarium*). The exception to this pattern was that the most dominant genus *Cladosporium* (OTU287) showed a positive relationship with Didymellaceae (Fig 3B).

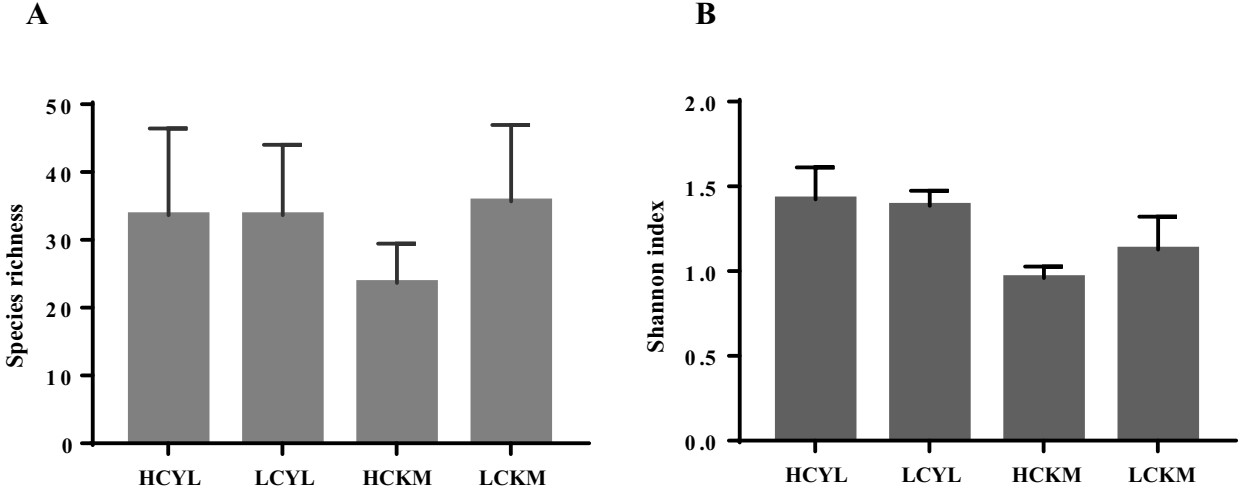

**Fig 1. Fungal alpha diversity.** LC: the air in the low part of the *A. adenophora* canopy; HC: the air in the high part of the *A. adenophora* canopy; YL: samples from Yunlong county (invaded region); and KM: samples from Kunming city (invaded region). The error bars represent one SE.

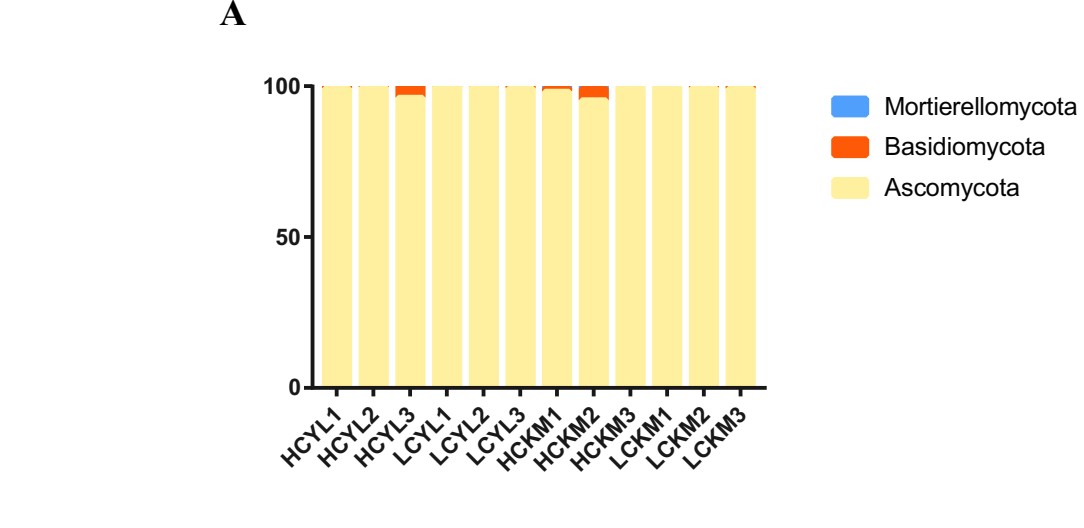

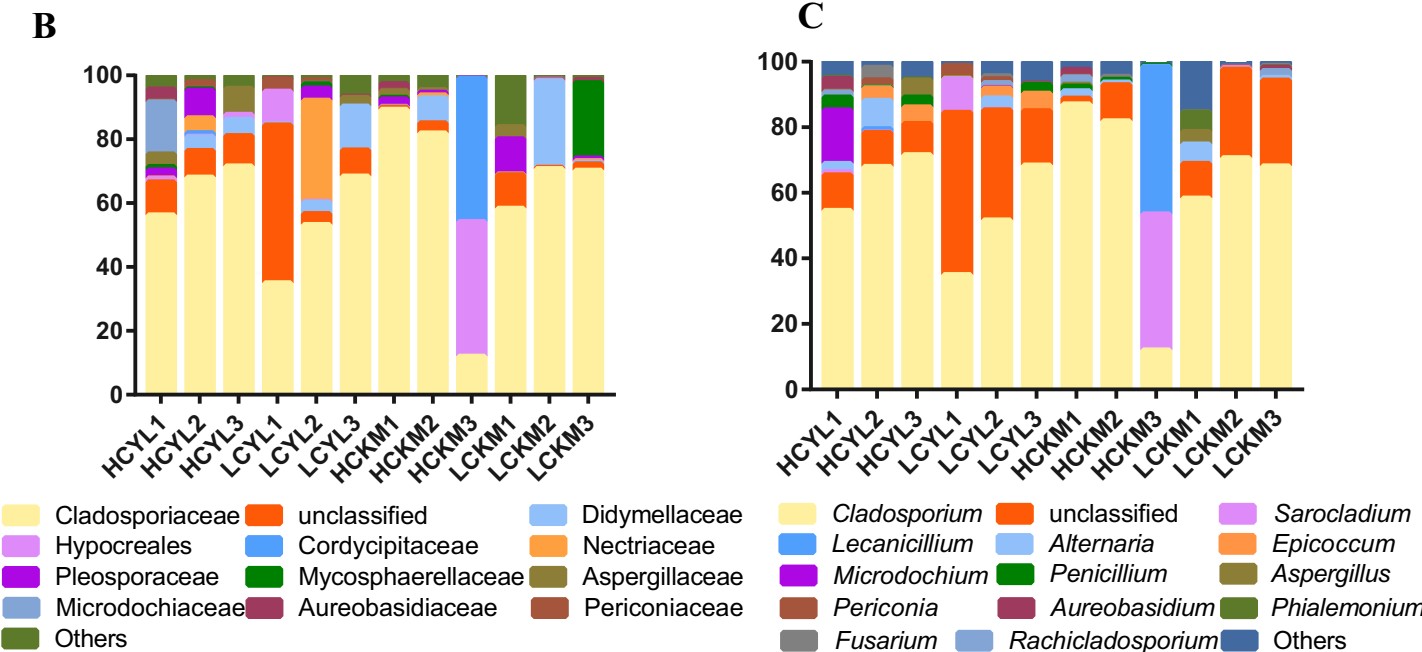

**Fig 2.** Relative abundances of the fungal distribution at the phyla (A), family (B), and genus level(C). Others in (B) and (C) represent groups with mean relative abundance less than 0.5%.

We identified nine Didymellaceae OTUs, with a sum of 7.53% of the sequences. Phylogenetically, OTU150 was combined with *Ascochyta* into one clade, and the remaining OTUs were close to *Phoma*, *Epiccoccum*, *Cumuliphoma* and *Didymella*, which were clustered into another clade (Fig 4A). Relatively LC had a greater abundance of the Didymellaceae family than that of HC, with no statistical significance (P > 0.05, paired t test). OTU11 and OTU90 were the predominant fungi and appeared in most samples. OTU150 only occurred in HCin-KM2, and OTU52 was detected in HCin-YL2. The remaining OTUs occurred in 2 to 8 samples (Fig 4C).

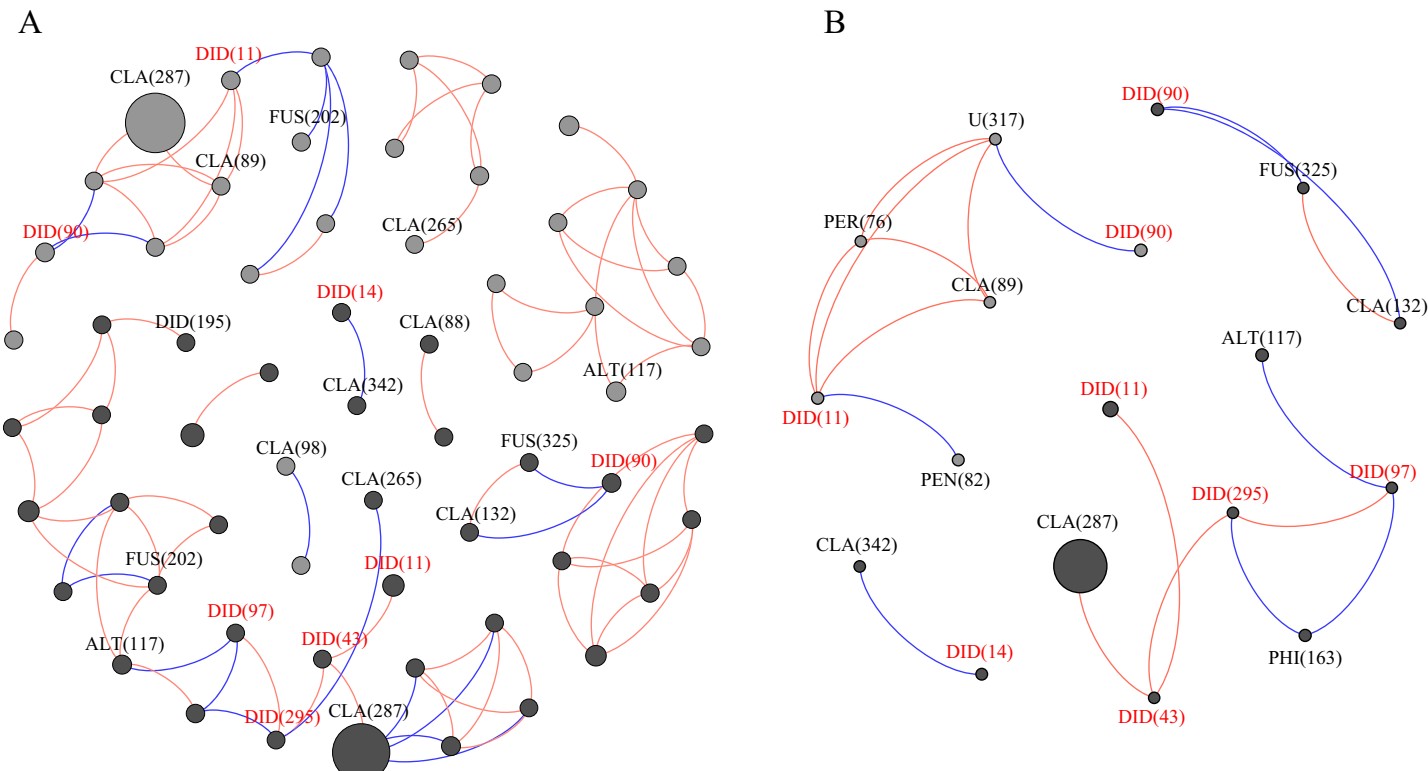

**Fig 3. Fungal co-occurrence networks.** (A) Total fungal co-occurrence network and (B) the co-occurrence networks between Didymellaceae and Didymellaceae-linked fungi. The gray and black nodes represent fungi in the HC and in LC, respectively. The node sizes are proportional to the OTU abundances, and the color of each line reflects positive (orange) or negative (blue) associations. The numbers in parentheses represent the OTU number. DID: Didymellaceae; PER: *Periconia*; PEN: *Penicillium*; CLA: *Cladosporium*; ALT: *Alternaria*; FUS: *Fusarium* and U: unclassified.

## Experiment 2: Isolation of fungi and disease experiment

**Fungal relationships between the canopy air and leaf spots of *A. adenophora*.**   The diverse Didymellaceae in the canopy air of *A. adenophora* may be associated with the easy release of Didymellaceae spores from monocultures of *A. adenopohra*. To verify whether *A. adenophora* harbors diverse Didymellaceae fungi, we selected one population, KM, to isolate the leaf spot fungi. In total, 355 isolates were isolated and grouped into 75 unique cOTUs with a 100% cutoff of ITS gene similarity. The most dominant fungi were from the family Glomerellaceae (24.51%). The second most dominant fungi were Didymellaceae, which included 10 cOTUs and accounted for 20.00% of the isolated strains (see S1 Table). By comparing the results with the fungal library obtained by high-throughput sequencing technology, a total of 20 overlapping OTUs were identified, which accounted for 11.45% and 44.79% of the abundance of two libraries, respectively (Fig 5A). The shared OTU11 (cOTU24), OTU14 (cOTU12, cOTU35, cOTU36, cOTU37 and cOTU43) and OTU295 (cOTU75) belonged to Didymellaceae were the most abundant fungi in both the canopy air and leaf spots. The leaf spots showed relatively high overlap with LC compared with that with HC (Fig 5B).

**Evaluation of the potential of Didymellaceae fungi to cause plant disease.**   Because Didymellaceae occurred frequently in both the canopy air and leaf spots of *A. adenophora*, 5 strains from the overlapped OTU14 were selected to test their virulence to native plants and *A. adenophora*. The disease experiments indicated that all the strains could infect *A. adenophora*. These strains also infected the 16 native plants tested to varying degrees. *A. adenophora* was more sensitive to these strains than most native plants, with the exception of *R. indica* (Fig 6).

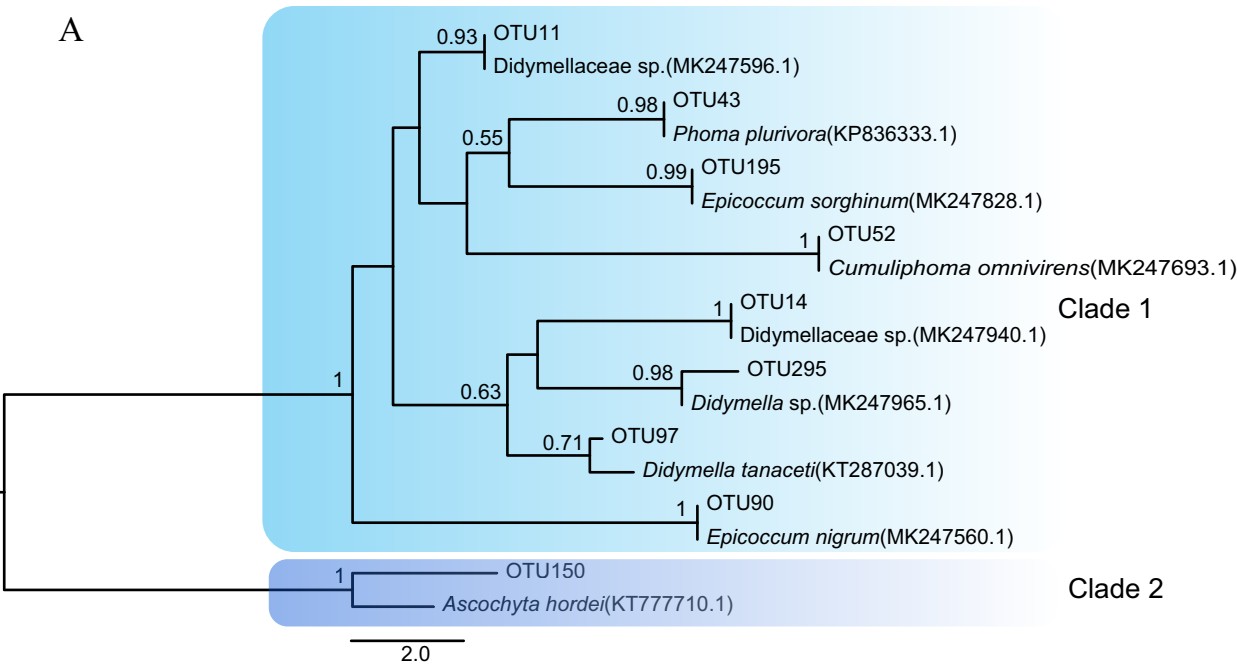

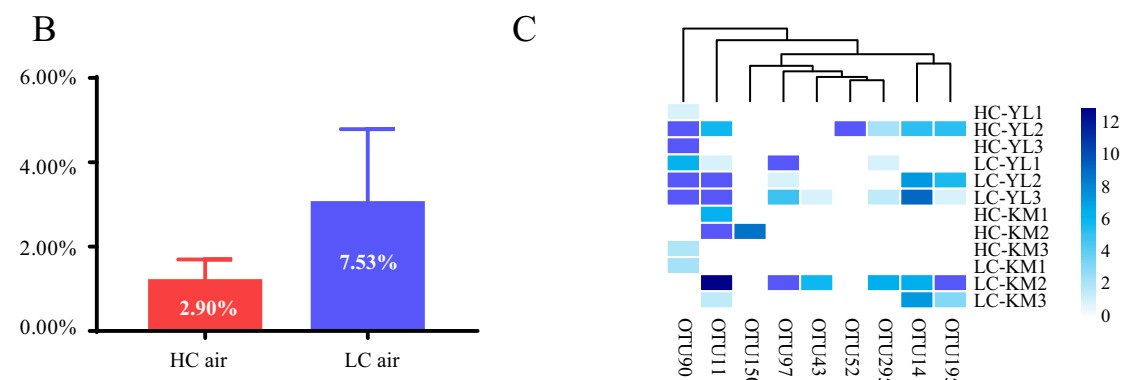

**Fig 4.** Phylogenetics(A),bar graph (B), and heatmap (C) of Didymellaceae. (A) The node stability was assessed with 1,000 bootstrap replicates, and only nodes with stability >0.5 are shown in the figure. The scale bar represents 2.0 substitutions per sequence position. (B)The percentage in the bar graph represents the percentage of Didymellaceae from different sources. The error bars represent one SE. (C)The relative abundances are expressed as the richness of fungi from a given isolation source transformed by $\log_2(x+1)$.

## Discussion

This is the first study that characterizes the airborne fungal communities of invasive plants and evaluates their potential to cause plant disease. Similar to previous reports, we found that Cladosporiaceae was the most frequent fungus in the air, mainly represented by the genus *Cladosporium* (Fig 2). Other fungi included common genera, such as *Fusarium*, *Alternaria* and *Epicoccum* [55–58]. Previously, there have been verified the occurrence of more fungal spores in the air in the low part of the canopy, which is close to plant leaves (i.e., the important release source of fungal spores), compared with the number of spores in the air of the high part of the canopy [20,59]. Several reports from crop systems have also verified this trend [27–29]. We did not found a higher fungal diversity and abundance in low part than high part of the canopy

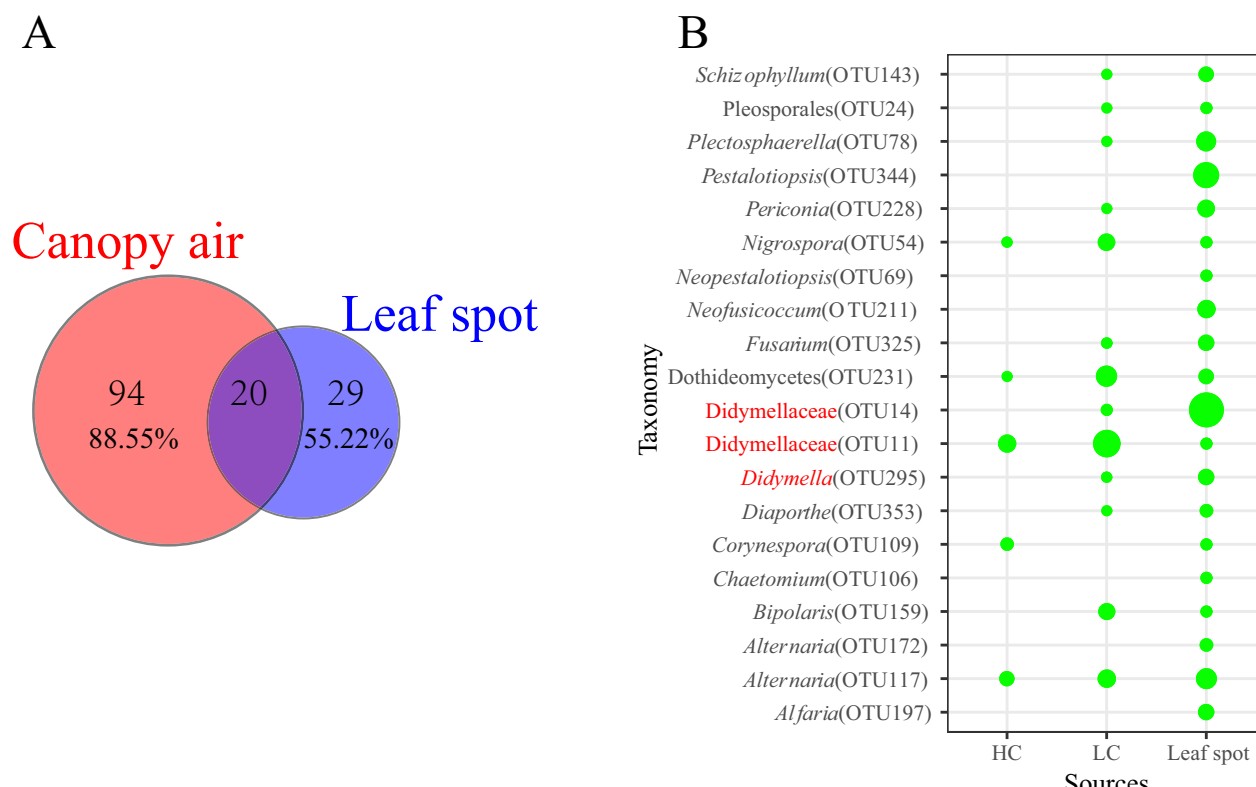

**Fig 5. Fungal overlap between the canopy air and leaf spots of *A. adenophora*.** (A) Venn diagram of the fungal overlap between the leaf spots and canopy air with a 97% similarity cutoff. The numbers above the diagram represent the number of OTUs. The percentages represent the relative abundances of the OTUs. (B) The bubble diagram of the relative abundances of overlapped OTUs. The size of the bubble is proportional to the relative abundance of the OTU. The OTU number in the parentheses represents the OTU number in the fungal library obtained by high-throughput sequencing technology (see S2 Table).

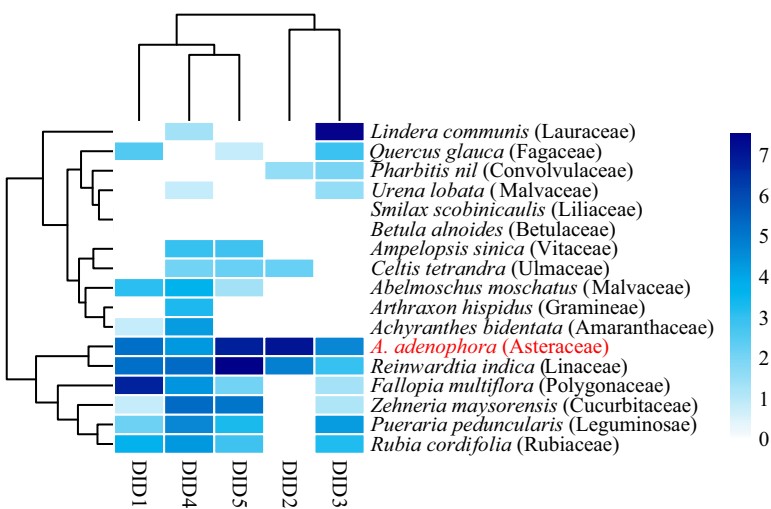

**Fig 6. Host range and virulence of five Didymellaceae fungal strains belonging to shared OTU14.** The relative abundances were expressed as the virulence, defined as leaf spot size (mm$^2$) transformed by log$_2$(x+1), of different fungi. All the fungi were isolated from leaf spots on *A. adenophora*.

(Fig 1). This is possible due to the small distance of two heights (~1.5m, see methods). Many reports have indicated that the determining factors for airborne pathogen spores are very complex, related to the environment, season and time [60–62], and in particular meteorological factors [21, 23, 55, 56, 63]. In this study, therefore, it is impossible to conclude the difference in fungal diversity across air column due to the small size of air samples (only two heights from two regions). However, partially supporting our expectation, we found that the air in the low part of the canopy (LC) had more fungal co-occurrences than the air in the high part of the canopy (HC) (Fig 3). In particular, we found that the dominant pathogens on *A*. *adenophora* enhanced the links among themselves (e.g., Didymellaceae fungi OTU90) and reduced their links with other fungi in the canopy air (Figs 3 and 4A).

This pattern may result from their high infection and easy release of spores from the monoculture *A*. *adenophora* host into the surrounding air. Species of Didymellaceae mainly cause leaf and stem lesions [64,65] and produce many small conidia, which is conducive for spreading through the air [66]. We also verified that *A*. *adenophora* harbored abundant leaf spot pathogens from Didymellaceae in this study (see S2 Table). The data suggest that there may form a feedback between the pathogenic fungi associated host and the airborne spores, i.e., the monoculture hosts support high load of pathogenic fungal infection, which will develop into a pathogenic-fungus-dominant canopy air and in turn further worsen their infection on hosts. Such a feedback cycle may partially contribute to the prevalence of Didymellaceae fungi on *A*. *adenophora*. Here we reasoned that the co-occurrence pattern of the dominant airborne fungal pathogen may be a label in the monoculture plant, including invasive plants, as well as economic crops. Janzen-Connell hypothesis indicates that diverse host-specific pathogen is important to maintain plant diversity [67, 68]. For native vegetation, higher plant diversity promotes higher diversity of fungal pathogens [69], it thus is expected that the airborne fungal co-occurrence become more abundant in native vegetation than in the monoculture plant. Although it remains to have such a conclusion due to the lack of air samples above native vegetation in this study, our data indirectly indicate that the diverse hosts and the diverse airborne pathogen co-occurrence network may partially contribute to high diversity of pathogens in native vegetation and thus decreases pathogen infection per plant [69]. Previously, increasing host diversity has been verified to help reduce the disease severity of airborne pathogens for monoculture crops. For example, Zhu *et al.* (2000) reported that planting disease-susceptible rice varieties in mixtures increased crop yield by 89% and reduced blast severity by 94% when comparing with those in monocultures [70]. Therefore, it is worthwhile to verify the commonness of such fungal co-occurrence patterns in other invasive systems as well as in crop systems, and the study of airborne fungal co-occurrence networks represents a promising field of plant pathology.

Regarding the reason why an exotic plant invades successfully, many reports have focused on the positive soil microbial feedback of invasive plants compared with that of native plants [8, 9]. Our data indicated that the pathogen feedback in the canopy air may weaken plant invasion as the residence time increases. In addition, previous reports have indicated that invasive plants could accumulate pathogens and infect native plants [10]. In this case, the infection risk was possible because susceptible native hosts were available, e.g., *R*. *indica*, in the invaded range (Fig 6). Nonetheless, whether these effects of Didymellaceae fungi ultimately translate into a competitive adverse of *A*. *adenophora* and the ecological risk must be evaluated against the background of the invaded ecosystems.

Interestingly, we found that the most dominant genus *Cladosporium* (OTU287) showed a positive relationship with Didymellaceae in the canopy air of *A*. *adenophora*. This pattern here does not mean that *Cladosporium* specifically facilitates the accumulation of Didymellaceae in the air but mirrors the high prevalence of *Cladosporium* spores in the air [55, 56]. In addition,

although Glomerellaceae fungi (e.g., *Colletotrichum*) were the most abundant fungi isolated from the leaf spots (also see Supplementary Data Fungi isolation table) as well as foliar endophytes of *A. adenophora* [71], they were rarely detected in the air by high-throughput sequencing. Because these *Colletotrichum* strains are rarely pathogenic on *A. adenophora* (unpublished data), we excluded them from this analysis. However, fungal spores in the air can be affected by microclimatic conditions, such as temperature and humidity [26], and it is worth determining the daily and seasonal dynamics of *Colletotrichum* in the air of *A. adenophora*.

For the fungi of Didymellaceae in the air, *Epicoccum* is the most reported genus [55], and the other groups have been disregarded because they were not identified. Didymellaceae is the largest family in the Pleosporales and has more than 5400 taxon names listed in MycoBank [72]. Although Chen *et al.* (2017) classify Didymellaceae into 19 genera, many Didymellaceae remain to be identified [66]. Similarly, most Didymellaceae fungi in this study were of the unclassified group, indicating that many of them may be potential novel species and worthy of characterization in the future. Nonetheless, the high-throughput sequences obtained from air samples we used for the comparison with the cultured strains were too short (~250bp) to fully confirm their matches. Therefore, the traditional culture methods should be used to successfully culture Didymellaceae fungi from the air, and both the morphological taxonomic methods and multiple loci of gene sequences are needed to accurately determine their phylogeny. Interestingly, strains from OTU14 showed a great variation in virulence and host range (Fig 6); in particular some strains, e.g., DID2, have a narrow range of hosts. These strains are the candidates to develop into a potential biocontrol of *A. adenophora* in future.

## Supporting information

**S1 Fig. Rarefaction curves.** LC, low part of canopy air of *A. adenphora* in the invaded region; HC, high part of canopy air of *A. adenphora* in the invaded region; YL, samples from Yunlong county (invaded region); KM, samples from Kunming city (invaded region).
(DOCX)

**S1 Table. Sampling sites information.**
(DOCX)

**S2 Table. Sub-sampled OTUs table and Fungi isolation table.**
(XLSX)

## Acknowledgments

The authors thank Jie Zhou, Zhi-Ping Yang, Tian Zeng, Li-Yuan Qin, and Wen-Ti Zheng at Yunnan University for help with sampling in the field and performing the disease experiment. Dr. Huan-Chong Wang and Dr. Tao Xu at Yunnan University helped with the identification of plant species.

## Author Contributions

**Conceptualization:** Lin Chen, Han-Bo Zhang.

**Data curation:** Lin Chen, Kai Fang, Xing-Fan Dong, Ai-Ling Yang, Yu-Xuan Li.

**Formal analysis:** Lin Chen, Han-Bo Zhang.

**Funding acquisition:** Han-Bo Zhang.

**Investigation:** Lin Chen, Kai Fang, Xing-Fan Dong, Ai-Ling Yang, Yu-Xuan Li.

**Methodology:** Lin Chen.

**Writing – original draft:** Lin Chen, Han-Bo Zhang.

**Writing – review & editing:** Lin Chen, Kai Fang, Xing-Fan Dong, Ai-Ling Yang, Yu-Xuan Li, Han-Bo Zhang.

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
