## [Decision Letter · Decision Letter 0]

7 Jan 2020

PONE-D-19-32348

Characterization of the fungal community in the canopy air of the invasive plant Ageratina adenophora and its potential to cause plant diseases

PLOS ONE

Dear Professor Zhang,

Thank you for submitting your manuscript to PLOS ONE. After careful consideration, we feel that it has merit but does not fully meet PLOS ONE’s publication criteria as it currently stands. Therefore, we invite you to submit a revised version of the manuscript that addresses the points raised during the review process.

The paper has been revised by two experts who suggested major revisions in order to improve the ms and render it acceptable for publication in Plos One.

We would appreciate receiving your revised manuscript by Feb 21 2020 11:59PM. To enhance the reproducibility of your results, we recommend that if applicable you deposit your laboratory protocols in protocols.io, where a protocol can be assigned its own identifier (DOI) such that it can be cited independently in the future. For instructions see: http://journals.plos.org/plosone/s/submission-guidelines#loc-laboratory-protocols

We look forward to receiving your revised manuscript.

Kind regards,

Sabrina Sarrocco

Academic Editor

PLOS ONE

Journal Requirements:

3. We note that you are reporting an analysis of a microarray, next-generation sequencing, or deep sequencing data set. PLOS requires that authors comply with field-specific standards for preparation, recording, and deposition of data in repositories appropriate to their field. Please upload these data to a stable, public repository (such as ArrayExpress, Gene Expression Omnibus (GEO), DNA Data Bank of Japan (DDBJ), NCBI GenBank, NCBI Sequence Read Archive, or EMBL Nucleotide Sequence Database (ENA)). In your revised cover letter, please provide the relevant accession numbers that may be used to access these data. For a full list of recommended repositories, see http://journals.plos.org/plosone/s/data-availability#loc-omics or http://journals.plos.org/plosone/s/data-availability#loc-sequencing.

Reviewers' comments:

Reviewer's Responses to Questions

**Comments to the Author**

1. Is the manuscript technically sound, and do the data support the conclusions?

Reviewer #1: Partly

Reviewer #2: Partly

2. Has the statistical analysis been performed appropriately and rigorously? 

Reviewer #1: No

Reviewer #2: Yes

3. Have the authors made all data underlying the findings in their manuscript fully available?

Reviewer #1: No

Reviewer #2: No

4. Is the manuscript presented in an intelligible fashion and written in standard English?

Reviewer #1: Yes

Reviewer #2: Yes

5. Review Comments to the Author

Reviewer #1: Thank you for the opportunity to review this paper, which focuses on documenting fungi represented as spores in the air column of an invasive plant, and then isolating fungi from that plant species and evaluating them for pathogenicity on native plants.

1. In some cases the current state of the field seems somewhat mis-represented. Airborne fungal propagules have been a major focus of epidemiological studies in plant pathology for some time. In other cases, the introduction sometimes is rather basic, making statements (for example about the disease triangle) that read as rather simplistic. I suggest that the authors revise the introduction to more appropriately integrate the strong traditions and current views of plant pathology.

2. The introduction might also be improved by improving synthesis; right now, many papers are listed and their results stated almost like a list, rather than in a synthetic, integrative way.

3. The two parts of the paper are somewhat disconnected and should be integrated more effectively. This reads as though it is two papers combined into one.

4. At times the methods lack key information. For example:

- The sample size and sampling approach should be clarified without requiring the reader to go to the supplement. From the main text it seem that the authors have sampled two regions with three sites per region. They sampled two heights in the air column. So, I think their study is based in this section on 12 samples (6 at each height). But then it is difficult to (1) understand why this sampling structure was chosen to address the research questions and (2) why there are four bars on the figures in Fig. 1 (it seems that this refers to 2 heights in 2 regions) -- but then, to what exactly are the statistics referring? Additional clarification will aid the reader greatly. At the moment the paper is weakened by the disconnect between questions, methods, and inference.

- What negative controls were used in the next-gen study? It is important that the extraction blanks, for example, be described.

- What positive controls did the authors use to ensure that their next-gen study generated data that could be used to estimate abundance, which is important for the two diversity indices? Also, why did the authors use two diversity indices - would it not be more fitting to use only species richness, if there is no positive control that validates read abundance?

- Were the next-gen data submitted to GenBank? This is stated for the cultures but not for the next-gen data.

- The comparison method for determining whether the ITS sequences matched for the air samples (next-gen) and cultures needs more careful attention. How were the comparisons made? Were deletions/insertions important or also gaps? Were composites from 97% similarity OTUs compared against the longer reads from cultures, and how, or were the comparisons done for all sequences?

- What controls were used in the inoculation experiments?

5. The presentation of the results leads to some concerns:

- The colors are often distracting in the figures, e.g., Fig 1.

- Exactly what is being tested with each statistical test is unclear. For example when lower canopy and higher canopy communities are compared it seems all of the data were thrown in, but wouldn't local differences be swamped by geographic/spatial differences? Shouldn't this be a paired test within sites, (low vs. high) X 3?

- When the authors use NMDS, what question do they seek to answer? Here again linking question with method with result will strengthen the paper. Were singletons removed from NMDS (and rare taxa appearing fewer times than the number of samples in the analysis)? What is the stress?

- The co-occurrence and taxonomic figures are really hard to see clearly. Is there a smart way to make them more useful?

- The authors identify taxa to genus based on very short fragments of a fast-evolving locus; some attention to limitations is needed to qualify their interpretation.

- The authors refer to five strains of OTU14 used in the inoculation experiments; were these truly five different strains? Is it possible this was exactly the same fungus isolated five times? More information to clarify will help. This is somewhat confusing because the authors refer to OTU14 but that OTU in the fungal isolation table is Xylariaceae.

- Is it likely that the growth form of the plants used in the experiment (Fig. 6) is important, our would family be more informative or useful to know?

5. The strong emphasis on the airborne fungi above an invasive plant being something special is weakened by the lack of air samples above native plants.

Reviewer #2: This paper describes the fungal composition of air at different heights of an invasive plant, in the context of how pathogen spread might impact further colonization of the plant and of native plants. Further information needs to be provided about the methods in order to interpret the results. The true impact of the findings on further plant invasion are speculative and should be presented as such. In addition, the idea that pathogens may foster diversity by inhibiting spread of plant progeny is known as the Janzen-Connell hypothesis. The authors could discuss their findings in the framework of the Janzen-Connell mechanism of maintaining plant diversity, particularly in the context of invasive species. Overall the paper presents the study in a clear way.

Particular comments:

Many claims need citations. For example, L44-46, did those studies show that the disease was spread via airborne propagules and not through other means? In L48-49, do references 9-10 show evidence that the spores spread over a long distance, rather than a short distance? L335, is there evidence that plant leaves and not the soil is the major release of fungal spores?

Why was it important to collect air at two heights? The implications of the findings about the composition of different heights was not discussed in terms of spread disease. What have other studies shown about compositional differences between ground level and 1.5m height?

Further information is needed about air sample collection. How many were taken at each site and when? Where they all collected in the same season? How long (minutes, hours) did sampling occur and at what time of day? Are the replicates biological or temporal?

L167: To what sequencing depth was each sample subsampled?

L228: it's unclear what a "heatmap of the leaf spots" is.

L247: How can community composition be compared with the Mann-Whitney U test? NMDS is a graphical tool and therefore cannot show significance.

Did the authors have any hypotheses about what they would find? Based on what we know about microbial communities in air, I would think the hypothesis would be that the effect of sampling site would be greater than sampling height; that, communities would cluster first by site, then by height.

Why were were the authors focused on Didymellaceae? This should be explained.

Co-occurence networks do not indicate interacting species. Rather, taxa could be co-occuring because of a common dispersal mechanism. The language should reflect this uncertainty associated with networks.

Editorial comments:

L44-46: odd punctuation

BLAST and UNITE should be capitalized

The authors present their data in the context of the maintaining plant diversity, but do not cite the Janzen-Connell hypothesis: https://en.wikipedia.org/wiki/Janzen%E2%80%93Connell_hypothesis

6. PLOS authors have the option to publish the peer review history of their article (what does this mean?). If published, this will include your full peer review and any attached files.

Reviewer #1: No

Reviewer #2: No

---

## [Author Response · Author response to Decision Letter 0]

18 Feb 2020

Response Letter

Reviewer #1: Thank you for the opportunity to review this paper, which focuses on documenting fungi represented as spores in the air column of an invasive plant, and then isolating fungi from that plant species and evaluating them for pathogenicity on native plants.

1. In some cases the current state of the field seems somewhat mis-represented. Airborne fungal propagules have been a major focus of epidemiological studies in plant pathology for some time. In other cases, the introduction sometimes is rather basic, making statements (for example about the disease triangle) that read as rather simplistic. I suggest that the authors revise the introduction to more appropriately integrate the strong traditions and current views of plant pathology.

Response: Thank you very much for your suggestions. As you pointed, it is somewhat mis-represented about introduction on current state of plant pathology. Many statements about airborne fungi are too basic and simplistic. By reorganizing the references, we narrowed these statements and focused the topic on the fungal pathology about invasive plants. Therefore, the introduction was rewritten. As you will see in this revision, our study aims were highlighted in the introduction and more supported the story as we showed in the title: to characterize the fungi associated with invasive plant. (Lines 46-106)

2. The introduction might also be improved by improving synthesis; right now, many papers are listed and their results stated almost like a list, rather than in a synthetic, integrative way.

Response: Thank you for your suggestion. As statements above, we improved the introduction in a synthetic integrative way. (Lines 46-106)

3. The two parts of the paper are somewhat disconnected and should be integrated more effectively. This reads as though it is two papers combined into one.

Response: Thank you for your suggestions. As statements above, we improved the introduction in a synthetic integrative way. In particular, we stated our study expectation in the introduction as: “It should be expected that: (1) The closer to the plant host, the higher diversity and more complicated co-occurring connections of fungi; (2) there are the sharing between fungi in the canopy air of invasive plants and host-associated pathogenic fungi. Currently, there is a total lack of the knowledge of the airborne phytopathogenic fungi surrounding the invasive plants.” (Lines 87-91) 

In this study, two experiments were designed to verify the expectation above. “Firstly, we used the high-throughput sequencing technology to characterize the fungal community and analyze their co-occurrence network in the canopy air of A. adenophora (experiment 1); and then the shared representative phytopathogenic fungi from family Didymellaceae were isolated from leaf spots of A. adenophora and their virulence to A. adenophora as well as native plants was tested (experiment 2).” (Lines 97-106)

In the methods and results, we divided the main content of this study into two experiments and added some sub-titles.

By this modification, it is better to understand what the story we would like to tell and why these two parts of study are designed. 

4. At times the methods lack key information. For example:

- The sample size and sampling approach should be clarified without requiring the reader to go to the supplement. From the main text it seem that the authors have sampled two regions with three sites per region. They sampled two heights in the air column. So, I think their study is based in this section on 12 samples (6 at each height). But then it is difficult to (1) understand why this sampling structure was chosen to address the research questions and (2) why there are four bars on the figures in Fig. 1 (it seems that this refers to 2 heights in 2 regions) -- but then, to what exactly are the statistics referring? Additional clarification will aid the reader greatly. At the moment the paper is weakened by the disconnect between questions, methods, and inference.

Response: In the method section, we added the information about sampling as “The samples were collected in two regions in which A. adenophora heavily occurs in Yunnan Province, China. These sampling sites are not located within the protected area and no sampling permit is required. At 2-4 pm, April 10 and 15, 2018, three sites per region were selected to collect the air in the same day and considered as the biological duplication. In each sampling site, we sampled two heights in the air column, and totally we collected 12 air samples.” (Lines 111-116)

And also the statistics were added as “The alpha fungal diversity between two regions, as well as between two heights in the air column was compared using paired t test by SPSS 25.0 (IBM, NY, USA). We used two methods to compare the diversity between two regions, i.e. considering the heights in the air column (LC: 3 vs 3, HC: 3 vs 3) and not considering the height in the air column (6 vs 6). The same method was used to compare the different height in the air column with and without considering the region.” Meanwhile, we revised the figure 1 as Shannon index and species richness in each sampling region and each heights in the air column.(Lines 183-193 and Fig. 1)

The reason for sampling structure was discussed in the revised manuscript as “Airborne fungal propagules have been a major focus of epidemiological studies in traditional plant pathology for native plants and crops. For example, most airborne spores come from fungi growing on living or dead plants [20], and their abundance is influenced by multiple abiotic factors, such as temperature [21], humidity [22], rainfall [23], and wind speed [24]. Scherm & van Bruggen (1995) reported that lesions on source plants infected with an isolate of Bremia lactucae sporulated at night and released spores beginning at sunrise [25]. Plant canopy air, with complex microclimates, thus acts as an important space for dynamic host-pathogen interactions [26]. Pathogens in the canopy air can infect plants under optimal microclimate conditions and in turn, release their spores into the air to complete their life cycle; meanwhile, the host can change the canopy microclimate to affect pathogen spread and disease development [26]. Some studies for the monoculture crop also have shown that numbers of conidia collected decreased with increasing height within and above the crop canopy [27-29]. Similarly, for the monoculture invasive plant, its canopy air would harbor diverse fungal spores released from leaf spots, which in turn, may infect host itself as well as neighboring species. It should be expected that: (1) The closer to the plant host, the higher diversity and more complicated co-occurring connections of fungi; (2) there are the sharing between fungi in the canopy air of invasive plants and host-associated pathogenic fungi. Currently, there is a total lack of the knowledge of the airborne phytopathogenic fungi surrounding the invasive plants.” (Lines 72-91)

- What negative controls were used in the next-gen study? It is important that the extraction blanks, for example, be described.

Response: Thank you for your concern. When we did the DNA extraction, a sterilized cellulose acetate membrane without collection of spores was used as the extraction blanks. The negative control which was failed for DNA extraction was excluded from the further processing. We added this information in the revised manuscript (Lines 140-142). 

- What positive controls did the authors use to ensure that their next-gen study generated data that could be used to estimate abundance, which is important for the two diversity indices? Also, why did the authors use two diversity indices - would it not be more fitting to use only species richness, if there is no positive control that validates read abundance?

Response: Thank you for your concern. As the text we mentioned “The extracted DNA was diluted to a concentration of 2 ng/μL to balance the amount of DNA among the samples” (Lines 138-139), and also “After OTU classification, we first deleted the unidentified OTUs at the phylum level and then subsampled the 30376 reads (minimized reads' sample) using the “sub.sample” function in MOTHUR v1.35.1. All the data were analyzed except those that were removed during subsampling (see S2 Table Data Subsampled OTUs table)” (Lines 176-180). And all these methods were used to ensure that the next-gen study generated data are rationale to estimate abundance across regions, or heights. The application of the two diversity indices is redundant, so in the revised version, only Shannon index and species richness were used for alpha diversity comparison.

- Were the next-gen data submitted to GenBank? This is stated for the cultures but not for the next-gen data.

Response: The next-gen data was submitted to GenBank under bioproject accession numbers PRJNA590841. (Lines 154-155)

- The comparison method for determining whether the ITS sequences matched for the air samples (next-gen) and cultures needs more careful attention. How were the comparisons made? Were deletions/insertions important or also gaps? Were composites from 97% similarity OTUs compared against the longer reads from cultures, and how, or were the comparisons done for all sequences? 

Response: Thank you very much. We answered your concerns by added the sentences in the methods as: “To identify the fungal sharing, the ITS sequences from each representative OTU obtained from airborne fungi were selected to perform alignment with those sequences obtained from cultural leaf spot fungi above. Because the ITS sequence obtained by the high-throughput sequencing technology was short (~250bp), the alignment was trimmed to this range and was clustered to generate OTUs again using MOTHUR software with a 97% similarity cutoff [43]. Deletions/insertions were considered when comparing the sequences from next-gen and cultures. Those OTUs clustered from both sources were defined as the shared fungi; the Venn and bubble diagrams were used to show the shared fungal OTUs.” (Lines 237-245)

- What controls were used in the inoculation experiments? 

Response: Thanks, we added the information of inoculation experiments as “The PDA culture agar without fungus was used as the negative control. In the case that the obvious symptom was developed in the negative control group, the same batch of the fungi was tested again.” (Lines 272-274)

5. The presentation of the results leads to some concerns:

- The colors are often distracting in the figures, e.g., Fig 1.

Response: Thanks. We changed color scheme. (Fig. 1)

- Exactly what is being tested with each statistical test is unclear. For example when lower canopy and higher canopy communities are compared it seems all of the data were thrown in, but wouldn't local differences be swamped by geographic/spatial differences? Shouldn't this be a paired test within sites, (low vs. high) X 3? 

Response: Thanks, as your concerns, we compared main taxonomies in different level (phylum, family, and genus) using a paired test within sites and also the heights in the air column. In the method section, we added the text as “The alpha fungal diversity between two regions, as well as between two heights in the air column was compared using paired t test by SPSS 25.0 (IBM, NY, USA). We used two methods to compare the diversity between two regions, i.e. considering the heights in the air column (LC: 3 vs 3, HC: 3 vs 3) and not considering the height in the air column (6 vs 6). The same method was used to compare the different height in the air column with and without considering the region. Similarly, the differences in fungal abundance between two regions, as well as between two heights in the air column were also compared using paired t test using the methods above described at phylum, family, and genus level, respectively. The fungi with an average abundance of less than 0.5% were excluded from the comparison.” (Lines 183-193)

In the result section, we also added the text as “The fungal community composition showed no significant difference between the LC and HC canopy air and between two regions (P > 0.05, paired t test), with the exception that Epicoccum varied significantly between YL and KM when no considering the LC and HC (P = 0.028, paired t test).” (Lines 299-302)

- When the authors use NMDS, what question do they seek to answer? Here again linking question with method with result will strengthen the paper. Were singletons removed from NMDS (and rare taxa appearing fewer times than the number of samples in the analysis)? What is the stress? 

Response: As the above paired t test used in the different level of the fungal community, there is no significant differences between region and the heights in the air column. And the NMDS analysis could not bring more information, so NMDS analysis was not used in the revised manuscript.

- The co-occurrence and taxonomic figures are really hard to see clearly. Is there a smart way to make them more useful? 

Response: Thank you for your concern. We redrew the co-occurrence network and we wish it is better to understand now. (Fig. 3).

- The authors identify taxa to genus based on very short fragments of a fast-evolving locus; some attention to limitations is needed to qualify their interpretation.

Response: Thank you very much for your suggestions. For the study of fungal diversity in next-generation sequencing, the classification of fungal groups using ITS fragments about 250 bp is a common method. However, the construction of phylogenetic trees using such short sequences is indeed largely limited. In the revised manuscript, we added the text in the discussion as “Nonetheless, the high-throughput sequences obtained from air samples we used for the comparison with the cultured strains were too short (~250bp) to fully confirm their matches. Therefore, the traditional culture methods should be used to successfully culture Didymellaceae fungi from the air, and both the morphological taxonomic methods and multiple loci of gene sequences are needed to accurately determine their phylogeny.” (Lines 468-476) 

- The authors refer to five strains of OTU14 used in the inoculation experiments; were these truly five different strains? Is it possible this was exactly the same fungus isolated five times? More information to clarify will help. This is somewhat confusing because the authors refer to OTU14 but that OTU in the fungal isolation table is Xylariaceae. 

Response: Thank you for your suggestions. The isolates we used in the disease experiment were isolated from different leaf spots. So even the DNA sequences of these isolates are the same, it is unlikely that they come from one clone. We added sentences like: “These cultured OTUs were named as cOTUs to distinguish from the OTUs generated by next-generation sequencing.” “The used multiple isolates from one OTU were obtained from different leaf spots to avoid the origin of the same clone.” (Lines 228-229 and 247-248 in methods )

And also in results, we added the text as “The shared OTU11 (cOTU24), OTU14 (cOTU12, cOTU35, cOTU36, cOTU37 and cOTU43) and OTU295 (cOTU75) belonged to Didymellaceae were the most abundant fungi in both the canopy air and leaf spots.” (Lines 359-362 in results)

- Is it likely that the growth form of the plants used in the experiment (Fig. 6) is important, our would family be more informative or useful to know?

Response: Thank you for concerns. From our current results the pathogens showed no preference neither on family nor on life form of plants. We deleted the description of life form in text, as well as in Fig. 6.

5. The strong emphasis on the airborne fungi above an invasive plant being something special is weakened by the lack of air samples above native plants.

Response: Thank you for your suggestions. As we reviewed in introduction, airborne fungi are influenced by multiple abiotic factors, such as temperature, humidity, rainfall, and wind speed. Therefore, to conclude the vegetation impact, a comparison of native vegetation vs invasive plant may need a large size of samples obtained from multiple invaded regions in order to the exclusion of the effects of these abiotic factors. This is already beyond the aim of current study. 

However, as suggested by this preliminary study, it is important and interesting to have such a comparison to understand disease epidemics and may provide new viewpoints for plant pathology. Therefore, we discussed this point in the discussion as: “This pattern may result from their high infection and easy release of spores from the monoculture A. adenophora host into the surrounding air. Species of Didymellaceae mainly cause leaf and stem lesions [64,65] and produce many small conidia, which is conducive for spreading through the air [66]. We also verified that A. adenophora harbored abundant leaf spot pathogens from Didymellaceae in this study (see S2Table). The data suggest that there may form a feedback between the pathogenic fungi associated host and the airborne spores, i.e., the monoculture hosts support high load of pathogenic fungal infection, which will develop into a pathogenic-fungus-dominant canopy air and in turn further worsen their infection on hosts. Such a feedback cycle may partially contribute to the prevalence of Didymellaceae fungi on A. adenophora. Here we reasoned that the co-occurrence pattern of the dominant airborne fungal pathogen may be a label in the monoculture plant, including invasive plants, as well as economic crops. Janzen-Connell hypothesis indicates that diverse host-specific pathogen is important to maintain plant diversity [67, 68]. For native vegetation, higher plant diversity promotes higher diversity of fungal pathogens [69], it thus is expected that the airborne fungal co-occurrence become more abundant in native vegetation than in the monoculture plant. Although it remains to have such a conclusion due to the lack of air samples above native vegetation in this study, our data indirectly indicate that the diverse hosts and the diverse airborne pathogen co-occurrence network may partially contribute to high diversity of pathogens in native vegetation and thus decreases pathogen infection per plant [69]. Previously, increasing host diversity has been verified to help reduce the disease severity of airborne pathogens for monoculture crops. For example, Zhu et al. (2000) reported that planting disease-susceptible rice varieties in mixtures increased crop yield by 89% and reduced blast severity by 94% when comparing with those in monocultures [70]. Therefore, it is worthwhile to verify the commonness of such fungal co-occurrence patterns in other invasive systems as well as in crop systems, and the study of airborne fungal co-occurrence networks represents a promising field of plant pathology. ” (Lines 409-437)

Reviewer #2: This paper describes the fungal composition of air at different heights of an invasive plant, in the context of how pathogen spread might impact further colonization of the plant and of native plants. Further information needs to be provided about the methods in order to interpret the results. The true impact of the findings on further plant invasion are speculative and should be presented as such. In addition, the idea that pathogens may foster diversity by inhibiting spread of plant progeny is known as the Janzen-Connell hypothesis. The authors could discuss their findings in the framework of the Janzen-Connell mechanism of maintaining plant diversity, particularly in the context of invasive species. Overall the paper presents the study in a clear way. 

Response: Thank you for your suggestions. As your concern, the impacts of the findings on further plant invasion are speculative, we added sentences in discussion as: “Nonetheless, whether these effects of Didymellaceae fungi ultimately translate into a competitive adverse of A. adenophora and the ecological risk must be evaluated against the background of the invaded ecosystems.” (Lines 445-447)

We also discussed our data in the frameworks of Janzen-Connell hypothesis as: “Janzen-Connell hypothesis indicates that diverse host-specific pathogen is important to maintain plant diversity [67, 68]. For native vegetation, higher plant diversity promotes higher diversity of fungal pathogens [69], it thus is expected that the airborne fungal co-occurrence become more abundant in native vegetation than in the monoculture plant. Although it remains to have such a conclusion due to the lack of air samples above native vegetation in this study, our data indirectly indicate that the diverse hosts and the diverse airborne pathogen co-occurrence network may partially contribute to high diversity of pathogens in native vegetation and thus decreases pathogen infection per plant [69]. Previously, increasing host diversity has been verified to help reduce the disease severity of airborne pathogens for monoculture crops. For example, Zhu et al. (2000) reported that planting disease-susceptible rice varieties in mixtures increased crop yield by 89% and reduced blast severity by 94% when comparing with those in monocultures [70]. Therefore, it is worthwhile to verify the commonness of such fungal co-occurrence patterns in other invasive systems as well as in crop systems, and the study of airborne fungal co-occurrence networks represents a promising field of plant pathology.” (Lines 421-437)

Particular comments:

Many claims need citations. For example, L44-46, did those studies show that the disease was spread via airborne propagules and not through other means? In L48-49, do references 9-10 show evidence that the spores spread over a long distance, rather than a short distance? L335, is there evidence that plant leaves and not the soil is the major release of fungal spores?

Response: Thank you very much for your suggestions. In this version, we rechecked our writing. We thought that there were somewhat mis-represented about the introduction on current state of plant pathology. Many statements about airborne fungi were too basic and simplistic. By reorganizing the references, we narrowed these statements and focused the topic on the fungal pathology about invasive plants. We rewrote the introduction. As you will see in this revision, our study aims were highlighted and more supported the story as we showed in title: to characterize the fungi associated with invasive plant. By doing this, the problems you mentioned here also were modified (Lines 46-106).

Why was it important to collect air at two heights? The implications of the findings about the composition of different heights was not discussed in terms of spread disease. What have other studies shown about compositional differences between ground level and 1.5m height?

Response: Thank you for your suggestions. 

In introduction, we added sentences to state why we collected airs in different heights as: “Plant canopy air, with complex microclimates, thus acts as an important space for dynamic host-pathogen interactions [26]. Pathogens in the canopy air can infect plants under optimal microclimate conditions and in turn, release their spores into the air to complete their life cycle; meanwhile, the host can change the canopy microclimate to affect pathogen spread and disease development [26]. Some studies for the monoculture crop also have shown that numbers of conidia collected decreased with increasing height within and above the crop canopy [27-29]. Similarly, for the monoculture invasive plant, its canopy air would harbor diverse fungal spores released from leaf spots, which in turn, may infect host itself as well as neighboring species. It should be expected that: (1) The closer to the plant host, the higher diversity and more complicated co-occurring connections of fungi; (2) there are the sharing between fungi in the canopy air of invasive plants and host-associated pathogenic fungi. Currently, there is a total lack of the knowledge of the airborne phytopathogenic fungi surrounding the invasive plants.” (Lines 78-91)

Again, we discussed this point in discussion section as: “Previously, there have been verified the occurrence of more fungal spores in the air in the low part of the canopy, which is close to plant leaves (i.e., the important release source of fungal spores), compared with the number of spores in the air of the high part of the canopy [20,59]. Several reports from crop systems have also verified this trend [27-29]. We did not found a higher fungal diversity and abundance in low part than high part of the canopy (Fig. 1). This is possible due to the small distance of two heights (~1.5m, see methods). Many reports have indicated that the determining factors for airborne pathogen spores are very complex, related to the environment, season and time [60-62], and in particular meteorological factors [21, 23, 55, 56, 63]. In this study, therefore, it is impossible to conclude the difference in fungal diversity across air column due to the small size of air samples (only two heights from two regions). However, partially supporting our expectation, we found that the air in the low part of the canopy (LC) had more fungal co-occurrences than the air in the high part of the canopy (HC) (Fig 3). In particular, we found that the dominant pathogens on A. adenophora enhanced the links among themselves (e.g., Didymellaceae fungi OTU90) and reduced their links with other fungi in the canopy air (Fig 3, 4A).” (Lines 392-408)

Further information is needed about air sample collection. How many were taken at each site and when? Where they all collected in the same season? How long (minutes, hours) did sampling occur and at what time of day? Are the replicates biological or temporal?

Response: We added the relative information in method section as “The samples were collected in two regions in which A. adenophora heavily occurs in Yunnan Province, China. These sampling sites are not located within the protected area and no sampling permit is required. At 2-4 pm, April 10 and 15, 2018, three sites per region were selected to collect the air in the same day and considered as the biological duplication. In each sampling site, we sampled two heights in the air column, and totally we collected 12 air samples. Briefly, in each region, three sampling sites over 1 km apart were randomly selected (S1 Table, sampling sites information). The canopy air in each site was sampled from both the immediate canopy (the air from the low part of the canopy, defined as LC) and 1.5 meters above the vegetation (the air from the high part of the canopy, defined as HC). For each sample, airborne spores were concentrated from 1000 liters of air (~5mins), using a 9cm diameter Petri dish filled with a sterilized cellulose acetate membrane by a surface air system (SAS) Super ISO 180 (VWR International PBI SRL, San Giusto, Italy).” (Lines 111-124)

L167: To what sequencing depth was each sample subsampled?

Response: Thank you for your concern. We added sentences in methods as: “After OTU classification, we first deleted the unidentified OTUs at the phylum level and then subsampled all samples to the 30376 reads (minimized reads' sample) using the “sub.sample” function in MOTHUR v1.35.1.” (Lines 176-178) 

L228: it's unclear what a "heatmap of the leaf spots" is.

Response: we changed this sentence to “One week after the inoculation, the leaves were harvested and leaf spot size was measured. The leaf spot size for each strain was visualized by a heat map, which was plotted by the R package "pheatmap".” (Lines 270-272)

L247: How can community composition be compared with the Mann-Whitney U test? NMDS is a graphical tool and therefore cannot show significance. 

Response: Thank you very much. We changed the Mann-Whitney U test to paired t test for comparison of fungal community structure. We added the relative information in method section as “The alpha fungal diversity between two regions, as well as between two heights in the air column was compared using paired t test by SPSS 25.0 (IBM, NY, USA). We used two methods to compare the diversity between two regions, i.e. considering the heights in the air column (LC: 3 vs 3, HC: 3 vs 3) and not considering the height in the air column (6 vs 6). The same method was used to compare the different height in the air column with and without considering the region. Similarly, the differences in fungal abundance between two regions, as well as between two heights in the air column were also compared using paired t test using the methods above described at phylum, family, and genus level, respectively. The fungi with an average abundance of less than 0.5% were excluded from the comparison.” (Lines 183-193)

As showed by the paired t test used in the different level of the fungal community, there was no significant difference between regions, and between the heights in the air column. And the NMDS analysis could not provide more information and was not used in the revised manuscript.

Did the authors have any hypotheses about what they would find? Based on what we know about microbial communities in air, I would think the hypothesis would be that the effect of sampling site would be greater than sampling height; that, communities would cluster first by site, then by height.

Response: Thank you for your suggestions. As you pointed and references reviewed in the introduction, airborne fungal spores are influenced by a lot of abiotic factors, in particular climate including temperature, wetness, etc. Therefore, as your hypothesis it would be that the effect of sampling site would be greater than sampling height; that, communities would cluster first by site, then by height. However, it failed to get such a conclusion when we used the paired t test for comparison of fungal community structure to test this hypothesis. Indeed, our current sample size (12 samples collected from two region) is not far enough to conclude the difference in fungal community composition between the regions, and between the heights in the air column in our study. Therefore, we changed the introduction (as you see statements above) to focus our aims on the characterization of fungal co-occurrence networks and fungal sharing between airs and host, as well as their potentials to be pathogenic. 

In introduction, we said: “It should be expected that: It should be expected that: (1) The closer to the plant host, the higher diversity and more complicated co-occurring connections of fungi; (2) there are the sharing between fungi in the canopy air of invasive plants and host-associated pathogenic fungi. Currently, there is a total lack of the knowledge of the airborne phytopathogenic fungi surrounding the invasive plants.” (Lines 87-91)

In results: “The alpha diversities, including the Shannon index and species richness, were different neither between the regions (with or without considering the height in the air column) nor between the high (HC) and low (LC) parts of the canopy (with or without considering the region) (Fig 1, P > 0.05, paired t test), but there was a large variation within sites.” (Lines 281-285)

In discussion: “Previously, there have been verified the occurrence of more fungal spores in the air in the low part of the canopy, which is close to plant leaves (i.e., the important release source of fungal spores), compared with the number of spores in the air of the high part of the canopy [20,59]. Several reports from crop systems have also verified this trend [27-29]. We did not found a higher fungal diversity and abundance in low part than high part of the canopy (Fig. 1). This is possible due to the small distance of two heights (~1.5m, see methods). Many reports have indicated that the determining factors for airborne pathogen spores are very complex, related to the environment, season and time [60-62], and in particular meteorological factors [21, 23, 55, 56, 63]. In this study, therefore, it is impossible to conclude the difference in fungal diversity across air column due to the small size of air samples (only two heights from two regions). However, partially supporting our expectation, we found that the air in the low part of the canopy (LC) had more fungal co-occurrences than the air in the high part of the canopy (HC) (Fig 3). In particular, we found that the dominant pathogens on A. adenophora enhanced the links among themselves (e.g., Didymellaceae fungi OTU90) and reduced their links with other fungi in the canopy air (Fig 3, 4A). 

This pattern may result from their high infection and easy release of spores from the monoculture A. adenophora host into the surrounding air. Species of Didymellaceae mainly cause leaf and stem lesions [64,65] and produce many small conidia, which is conducive for spreading through the air [66]. We also verified that A. adenophora harbored abundant leaf spot pathogens from Didymellaceae in this study (see S2Table). The data suggest that there may form a feedback between the pathogenic fungi associated host and the airborne spores, i.e., the monoculture hosts support high load of pathogenic fungal infection, which will develop into a pathogenic-fungus-dominant canopy air and in turn further worsen their infection on hosts. Such a feedback cycle may partially contribute to the prevalence of Didymellaceae fungi on A. adenophora. Here we reasoned that the co-occurrence pattern of the dominant airborne fungal pathogen may be a label in the monoculture plant, including invasive plants, as well as economic crops. Janzen-Connell hypothesis indicates that diverse host-specific pathogen is important to maintain plant diversity [67, 68]. For native vegetation, higher plant diversity promotes higher diversity of fungal pathogens [69], it thus is expected that the airborne fungal co-occurrence become more abundant in native vegetation than in the monoculture plant. Although it remains to have such a conclusion due to the lack of air samples above native vegetation in this study, our data indirectly indicate that the diverse hosts and the diverse airborne pathogen co-occurrence network may partially contribute to high diversity of pathogens in native vegetation and thus decreases pathogen infection per plant [69]. Previously, increasing host diversity has been verified to help reduce the disease severity of airborne pathogens for monoculture crops. For example, Zhu et al. (2000) reported that planting disease-susceptible rice varieties in mixtures increased crop yield by 89% and reduced blast severity by 94% when comparing with those in monocultures [70]. Therefore, it is worthwhile to verify the commonness of such fungal co-occurrence patterns in other invasive systems as well as in crop systems, and the study of airborne fungal co-occurrence networks represents a promising field of plant pathology.” (Lines 392-437)

Why were were the authors focused on Didymellaceae? This should be explained.

Response: Thank you for your concerns. Our team has studied the fungi (including endophytes and pathogens) associated with A. adenophora for a long time. In two papers we published very recently, we found foliar Didymellaceae fungi are adverse to the growth of A. adenophora and therefore are considered as potential leaf spot pathogens; again, we also verified the occurrence of this family of fungi in the surrounding air of A. adenophora. Therefore we focused on these fungi in this paper. 

Indeed, our results in this paper showed that these fungi are dominant on the leaf spot pathogens of A. adenophora, and they were also distributed frequently in the surrounding air.

As your concerns, we cited our previous findings and added sentences in the introduction as: “In recent, our group has indicated that the foliar fungi from family Didymellaceae are adverse to the growth of A. adenophora [36], and these fungi frequently occurs in the surrounding environment, such as in the withered leaves and the canopy air of hosts [37].” (Lines 97-100)

Co-occurence networks do not indicate interacting species. Rather, taxa could be co-occuring because of a common dispersal mechanism. The language should reflect this uncertainty associated with networks.

Response: Yes, we agreed with your point. The co-occurrence network represents a common dispersal mechanism. We deleted all words “interaction” and such meanings when describing the co-occurrence networks. And we added some information about co-occurrence in the methods section as “Co-occurrence patterns do not allow mapping of microbial interactions directly, but provide information on particular groups sharing habitats or performing similar ecological functions [46].” (Lines 200-202)

In particular, we added sentences in the discussion as:

“This pattern may result from their high infection and easy release of spores from the monoculture A. adenophora host into the surrounding air. Species of Didymellaceae mainly cause leaf and stem lesions [64,65] and produce many small conidia, which is conducive for spreading through the air [66]. We also verified that A. adenophora harbored abundant leaf spot pathogens from Didymellaceae in this study (see S2Table). The data suggest that there may form a feedback between the pathogenic fungi associated host and the airborne spores, i.e., the monoculture hosts support high load of pathogenic fungal infection, which will develop into a pathogenic-fungus-dominant canopy air and in turn further worsen their infection on hosts. Such a feedback cycle may partially contribute to the prevalence of Didymellaceae fungi on A. adenophora.” (Lines 409-419)

“Interestingly, we found that the most dominant genus Cladosporium (OTU287) showed a positive relationship with Didymellaceae in the canopy air of A. adenophora. This pattern here does not mean that Cladosporium specifically facilitates the accumulation of Didymellaceae in the air but mirrors the high prevalence of Cladosporium spores in the air [55, 56].” (Lines 448-452)

Editorial comments:

L44-46: odd punctuation

Response: Thank you very much for your suggestions. Based on two reviewer’s suggestions, we reorganized the references and rewrote our introduction to highlight our study aim. No such a problem existed in this version.

BLAST and UNITE should be capitalized

Response: Thank you editor. We corrected it.

The authors present their data in the context of the maintaining plant diversity, but do not cite the Janzen-Connell hypothesis: https://en.wikipedia.org/wiki/Janzen%E2%80%93Connell_hypothesis

Response: Thanks you very much for your suggestions. We cited the reference and also discussed our data in the frameworks of Janzen-Connell hypothesis as: “Janzen-Connell hypothesis indicates that diverse host-specific pathogen is important to maintain plant diversity [67, 68]. For native vegetation, higher plant diversity promotes higher diversity of fungal pathogens [69], it thus is expected that the airborne fungal co-occurrence become more abundant in native vegetation than in the monoculture plant. Although it remains to have such a conclusion due to the lack of air samples above native vegetation in this study, our data indirectly indicate that the diverse hosts and the diverse airborne pathogen co-occurrence network may partially contribute to high diversity of pathogens in native vegetation and thus decreases pathogen infection per plant [69]. Previously, increasing host diversity has been verified to help reduce the disease severity of airborne pathogens for monoculture crops. For example, Zhu et al. (2000) reported that planting disease-susceptible rice varieties in mixtures increased crop yield by 89% and reduced blast severity by 94% when comparing with those in monocultures [70]. Therefore, it is worthwhile to verify the commonness of such fungal co-occurrence patterns in other invasive systems as well as in crop systems, and the study of airborne fungal co-occurrence networks represents a promising field of plant pathology.” (Line 421-437)

---

## [Decision Letter · Decision Letter 1]

10 Mar 2020

Characterization of the fungal c ommunity in the canopy air of the invasive plant Ageratina adenophora and its potential to cause plant diseases

PONE-D-19-32348R1

Dear Dr. Zhang,

We are pleased to inform you that your manuscript has been judged scientifically suitable for publication and will be formally accepted for publication once it complies with all outstanding technical requirements.

With kind regards,

Sabrina Sarrocco

Academic Editor

PLOS ONE

Additional Editor Comments (optional):

Reviewers' comments:

Reviewer's Responses to Questions

**Comments to the Author**

1. If the authors have adequately addressed your comments raised in a previous round of review and you feel that this manuscript is now acceptable for publication, you may indicate that here to bypass the “Comments to the Author” section, enter your conflict of interest statement in the “Confidential to Editor” section, and submit your "Accept" recommendation.

Reviewer #2: All comments have been addressed

2. Is the manuscript technically sound, and do the data support the conclusions?

Reviewer #2: (No Response)

3. Has the statistical analysis been performed appropriately and rigorously? 

Reviewer #2: (No Response)

4. Have the authors made all data underlying the findings in their manuscript fully available?

Reviewer #2: (No Response)

5. Is the manuscript presented in an intelligible fashion and written in standard English?

Reviewer #2: (No Response)

6. Review Comments to the Author

Reviewer #2: (No Response)

7. PLOS authors have the option to publish the peer review history of their article (what does this mean?). If published, this will include your full peer review and any attached files.

Reviewer #2: No

---

## [Editor Report · Acceptance letter]

12 Mar 2020

PONE-D-19-32348R1 

Characterization of the fungal community in the canopy air of the invasive plant *Ageratina adenophora* and its potential to cause plant diseases 

Dear Dr. Zhang:

I am pleased to inform you that your manuscript has been deemed suitable for publication in PLOS ONE. Congratulations! Your manuscript is now with our production department. 

With kind regards,

on behalf of

Dr Sabrina Sarrocco 

Academic Editor

PLOS ONE